# Residual Stresses Induced by Surface Working and Their Improvement by Emery Paper Polishing

**Makoto Hayashi** †

Hitachi, Ltd., 1-1, Saiwai-cho, 3-chome, Hitachi, Ibaraki 317-0017, Japan; zv-hayashimak@aida.co.jp or hayashijapan7@msn.com; Tel.: +81-42-772-5231
† Now at AIDA ENGINEERING, LTD.

**Abstract:** In many of machine parts and structural components, materials surface would be worked. In this study, residual stresses on the surfaces were measured by X-ray diffraction method, and effects of surface working on the residual stresses were examined. In case of lathe machining of type 304 stainless steel bar, the residual stresses in circumferential directions are tensile, and those in axial directions are almost compressive. Highly tensile residual stresses in the circumferential directions were improved by emery paper polishing. 10 to 20 times of polishing changes high tensile residual stresses to compressive residual stresses. In the case of shot peening on a type 304 stainless steel plate, the compressive residual stress inside is several hundred MPa lower than that on the surface. By applying the emery paper polishing to the shot peened surface 10 or 20 times, the residual stress on the surface is improved to −700 MPa. While fatigue strength at 288 °C in the air of the shot peened material is 30 MPa higher than solution heat treated and electro-polished material, the fatigue strength of the shot peened and followed by emery paper polished material is 60 MPa higher. Thus, the emery paper polishing is simple and a very effective process for improvement of the residual stresses.

**Keywords:** residual stress; X-ray diffraction; lathe machining; shot peening; emery paper polishing; austenitic stainless steel; fatigue strength

## 1. Introduction

Several kinds of surface working such as turning, forging, heat-treatment, or surface finishing are applied to structural components and machine parts. In accordance with such working and heat treatment, residual stresses and plastic deformation are introduced in the material [1–4]. Compressive residual stress and work hardening induced by the plastic deformation increase fatigue strength, but tensile residual stress decreases it. Thus, shot peening, laser shock peening, and ultrasonic peening using steel wire that enables formation of the compressive residual stresses are applied to the structural components [5–8].

The residual stresses introduced by the surface working are affected by material itself and working conditions, so that the residual stresses on the surface and distributions in the sub-surface layer are measured using conventional X-ray, synchrotron radiation X-ray, and neutron diffraction [1,9]

On the other hand, the effect of pre-working on fatigue strength was discussed in terms of residual stress and hardness in order to separate their effects [10]. Furthermore, changes of residual stresses induced by turning under cyclic stress have been investigated [11–14].

However, all of these results have been mainly obtained using carbon steels. In light water reactors like boiling water reactor (BWR) and pressurized water reactor (PWR), several kinds of austenitic stainless steels are widely used. The effect of working on the residual stresses in austenitic stainless steel has not been studied. Only changes in residual stress in the worked surface layer of type 304 stainless steel due to tensile deformation has been studied [15].

The turning process commonly causes very high tensile residual stresses. These high tensile residual stresses adversely affect the fatigue strength or stress corrosion cracking in structural components. In this study, effects of turning and shot-peening on the residual stresses and their improvement by emery paper polishing were studied using the X-ray diffraction method.

## 2. Experimental Procedure

### 2.1. Materials

Materials used are type 304 stainless steel and XM19 (equivalent to ASTM A240). Chemical compositions and mechanical properties at room temperature and 288 °C are listed in Tables 1 and 2.

**Table 1.** Chemical composition (wt%).

| Material | Shape | C | Si | Mn | P | S | Ni | Cr | Mo | Nb | V | N |
|---|---|---|---|---|---|---|---|---|---|---|---|---|
| Type 304 | Bar | 0.05 | 0.58 | 0.95 | 0.3 | - | 9.01 | 18.2 | - | - | - | - |
| | Plate | 0.06 | 0.56 | 0.92 | 0.25 | - | 8.85 | 18.1 | - | - | - | - |
| XM19 (ASTM A240) | Ring | 0.05 | 0.72 | 5.2 | 0.03 | 0.02 | 12.1 | 22.1 | 2.1 | 0.25 | 0.16 | 0.32 |

**Table 2.** Mechanical properties.

| Material | Temperature (°C) | 0.2% Flow Stress (MPa) | Tensile Strength (MPa) | Elongation (%) | Reduction of Area (%) |
|---|---|---|---|---|---|
| Type 304 | RT | 260 | 573 | 59.7 | 81.7 |
| | 288 | 162 | 444 | 48.3 | 78.4 |
| XM19 (ASTM A240) | RT | 527 | 866 | 37.0 | - |

### 2.2. Surface Working

Twenty-five millimeters in diameter and 60 mm long round bar like specimens were prepared for the turning process (lathe machining). The round bar specimens were chucked in the lathe machine and their surfaces were worked by changing cutting speed, cutting depth, and feed speed. Tool material was conventional carbon tool steel. Cutting oil was not used.

### 2.3. Residual Stress Measurement

Surface residual stresses were measured by the conventional X-ray diffraction method. X-ray conditions are shown in Table 3. The measurement method of residual stress is referred from standards for X-ray stress measurement (2002 Version) [16]. Characteristic X-ray was Cr-Kβ. Diffraction plane was (311). Parallel beam slits were employed for incident and diffracted beams. Irradiated area was $4 \times 5$ mm$^2$ masked by vinyl tape. The residual stresses were determined by side inclination and sin$^2\psi$ method.

**Table 3.** X-ray conditions for residual stress measurement.

| Characteristic X-ray | Cr-Kβ |
|---|---|
| Diffraction plane | 311 |
| Tube voltage | 30 kV |
| Tube current | 10 mA |
| Scanning speed | 2°/min |
| Scanning angle range | 144–154° |
| Step angle | 0.3° |
| Integral time | 9 s |
| Divergent angle | 0.8° |
| Iirradiated area | $4 \times 5$ mm$^2$ |

The residual stress distributions in the sub-surface were measured by sequential polishing method using electrochemical etching liquid.

Incident X-ray beam is decayed by absorption. Penetration depth is defined as depth at which the intensity of X-rays entering the material is reduced to 1/e of the incident X-ray intensity on the surface. The penetration depth in the austenitic stainless steel using Cr-K$\beta$ is estimated as 7.1 $\mu$m.

### 2.4. Fatigue Test

Shape and dimensions of fatigue test specimen are shown in Figure 1. Minimum diameter of hour-glass type round specimen is 8 mm. Surface of specimens was finished with #240 emery paper after the lathe machining.

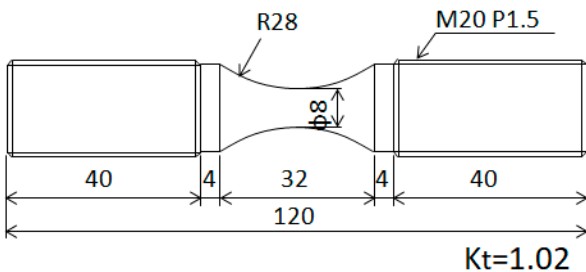

**Figure 1.** Shape and dimensions of fatigue test specimen.

Load controlled fatigue tests were performed at 288 °C in air. Stress ratio was −1. Frequency for cyclic loading was basically 1 Hz because, if the frequency is set at about 20 Hz from the beginning, the specimen is heated up and appropriate fatigue lives cannot be obtained. Thus, it is necessary for the specimen to be slowly work-hardened by low frequency stress cycling, whereas the frequency was gradually increased from 2 to 5, 10, 15, and 20 Hz after about $2 \times 10^4$ cycles for the specimen with estimated fatigue lives longer than $10^5$ cycles.

## 3. Experimental Results and Discussion

### 3.1. Residual Stress Induced by Lathe Machining

The residual stress is determined by the X-ray diffraction $\sin^2\psi$ method. In many of the ground or lathe machined metallic surfaces, the $\sin^2\psi$ diagram is not straight and so-called $\psi$-split takes place [17,18]. However, in these experiments, the $\psi$-split was not observed and $\sin^2\psi$ diagram is almost straight. Thus, the residual stresses were determined by the conventional $\sin^2\psi$ method.

The effect of cutting depth on the residual stress is shown in Figure 2. The cutting depth was changed from 0.1 mm to 1.0 mm. Axial residual stress ranges from about 0 MPa to −150 MPa but remains at around −100 MPa. Circumferential residual stress is tensile. It takes the maximum value for the cutting depth of 0.1 mm and then it gradually decreases with the cutting depth.

The effect of cutting speed on the residual stress is shown in Figure 3. The cutting speed was changed from 2 m/min to 95 m/min. The axial residual stress is kept almost compressive. The absolute value is small and −140 MPa at the maximum. The circumferential residual stress is tensile, it increases with the cutting speed, and takes the maximum value of 570 MPa for the cutting speed of 95 m/min.

The effect of feed on the residual stress is shown in Figure 4. Feed for every revolution of specimen was changed from 0.045 m/rev to 0.157 mm/rev. The axial residual stress is −130 MPa for the feed of 0.045 mm/rev. It slightly decreases with the feed but gradually increases with the feed. It turns tensile at the feed of 0.135 mm/rev and takes the maximum value of 80 MPa for the feed of 0.157 mm/rev. The circumferential residual stress is tensile, and it takes 340 MPa for the feed of 0.045 mm/rev. It gradually increases with the feed and takes the maximum value of 520 MPa for the feed of 0.157 mm/rev.

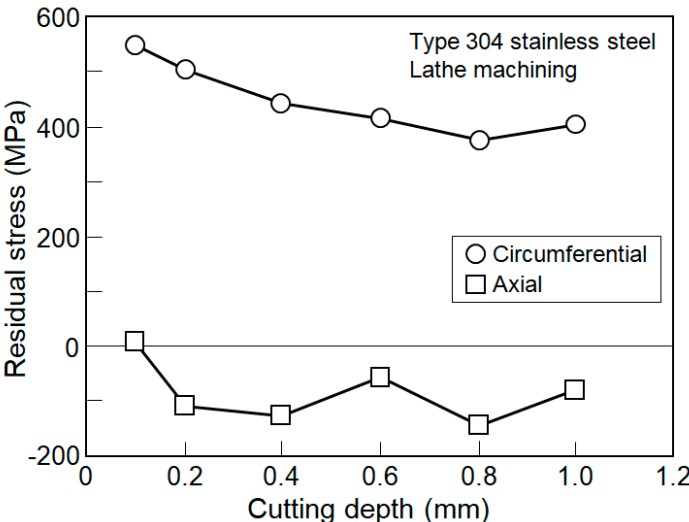

**Figure 2.** Effect of cutting depth on the residual stress.

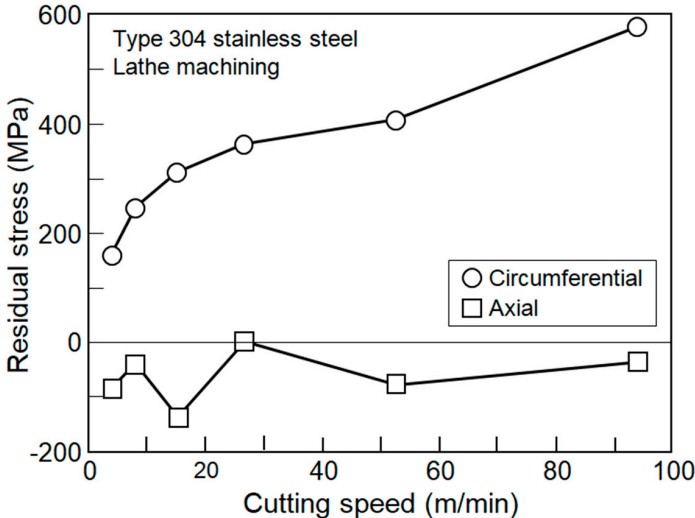

**Figure 3.** Effect of cutting speed on the residual stress.

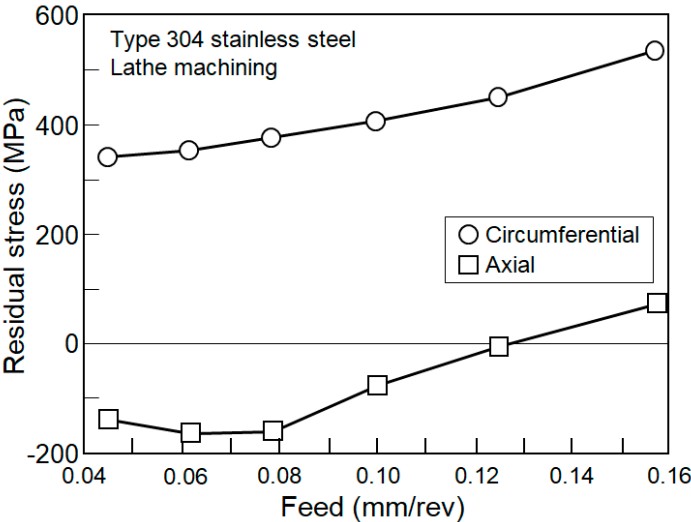

**Figure 4.** Effect of feed on the residual stress.

The residual stresses due to the lathe machining can be caused by two mechanisms. One is the burnishing effect [19]. The machined surface is cut by the tool materials and simultaneously compressed. This causes the compressive residual stress. On the other hand, the surface layer is cut away and plastically deformed. Thus, the surface layer is heated up, so that the surface layer is thermo-plastically deformed, especially in the austenitic stainless steel. This causes the tensile residual stress. In the axial direction, the burnishing effect exceeds and the compressive residual stress seems to be caused. In the circumferential direction, the thermo-plastically deformation exceeds and the tensile residual stress seems to be caused. In particular, not using cutting oil encourages this tendency.

The residual stresses introduced by the turning depend on various cutting condition. Thus, no generalized or unified interpretation has been made. Even though, the residual stresses in mild steel turned without cutting oil were measured by Yokoyama et al. [3]. The axial residual stress was compressive and −700 MPa. The circumferential residual stress was compressive and −500 MPa. The results of the effect of cutting speed show that the axial residual stress ranges from −400 MPa and −600 MPa, taking the compressive maximum value for the cutting speed of 100 mm/min. The circumferential residual stress ranges from −500 MPa and −200 MPa, and the compressive residual stress decreases with the increasing of cutting speed.

Results of the effect of feed show that the axial residual stress is −600 MPa for the feed of 0.03 mm/rev, but the compressive residual stress decreases with the increasing of feed; finally, it becomes −220 MPa for the feed of 0.45 mm/rev. The circumferential residual stress is −200 MPa for the feed of 0.03 mm/rev, but it increases with increasing of the feed; finally, it approaches −700 MPa for the feed of 0.45 mm/rev.

Results of the effect of cutting depth show that the axial residual stress gradually increases from −400 MPa and saturates at −600 MPa. The circumferential residual stress takes −150 MPa for small cutting depth, suddenly increases to −550 MPa, and finally gradually increases and saturates at −600 MPa.

In the carbon steel, the residual stresses induced by the turning are compressive both in the axial and circumferential direction [3,4,11]. On the other hand, in this experiment using austenitic stainless steel, the residual stresses are compressive in the axial direction but tensile in the circumferential direction. In the stainless steel, the work hardening rate is very much higher and the heat conductive coefficient is very much lower compared with the carbon steel. These two physical properties induce the thermo-plastic deformation at the tip of tool material and finally cause very much high tensile residual stresses in the circumferential direction.

The tensile residual stress adversely affects the fatigue strength or the stress corrosion cracking. Thus, emery paper polishing is subjected to the worked surface with tensile residual stress. Improvement of tensile residual stress by the emery paper polishing is shown in Figure 5. For the as-lathe machined condition, the residual stress is about 410 MPa. By the emery paper polishing, the tensile residual stress rapidly decreases and it turns compressive by about 100 polishings. The specimen is held in the chuck of lathe machine and the strip shaped emery paper is gripped at the turret. Thus, the polishing process is very easy. At the manufacturing site of actual equipment, the use of the flapper wheel seems to be much more cost and time effective.

The residual stress distribution in the sub-surface of lathe machined XM19 (equivalent to ASTM A240) ring is shown in Figure 6. Outside diameter of XM19 ring is 70 mm and thickness is 6 mm. The residual stresses were measured four times. The average residual stress on the surface is 920 MPa. This is due to very high yield stress (0.2% flow stress) and work hardening characteristics of XM19. The residual stress rapidly decreases with the depth and turns out to be compressive at about 20 µm. It takes the minimum value of about −250 MPa at the depth of 40 µm and gradually approaches zero.

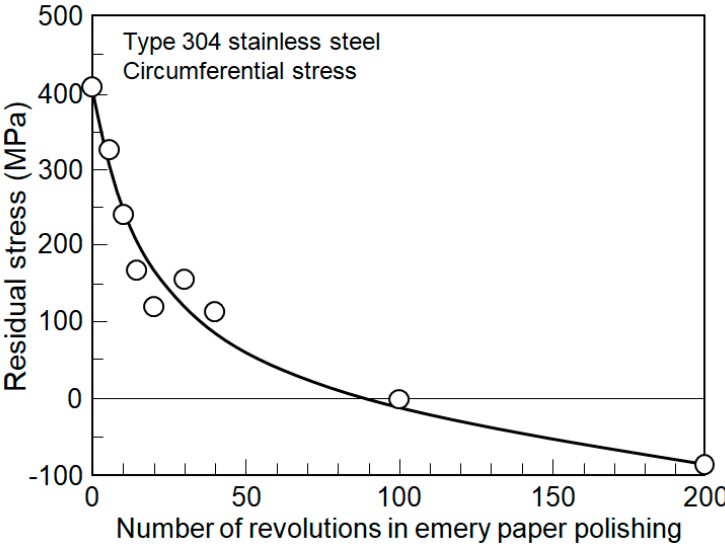

**Figure 5.** Improvement of tensile residual stress by emery paper polishing in lathe machined type 304 stainless steel.

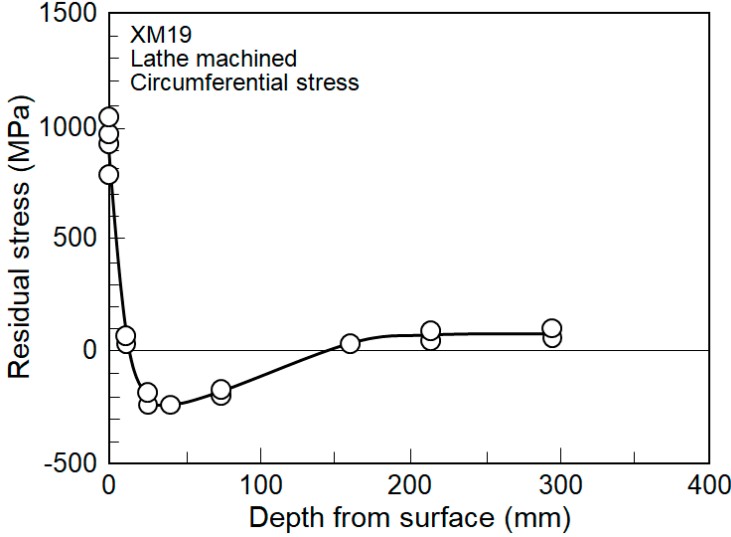

**Figure 6.** Circumferential residual stress distribution in lathe machined XM19 ring.

Distribution of half value width Hw is shown in Figure 7. The half value width for the solid solution heat treatment condition is about 1.1 degrees, as can be seen at deeper than 100 μm in the figure. The half value width at the surface is about 3.75 degrees. This means that the half value width at the surface is very much larger than that of the solid solution heat treatment condition and indicates that the surface layer is remarkably plastically deformed. This large plastic deformation seems to cause very high tensile residual stress. The half value width rapidly decreases with the depth. The distribution of half value width indicates that the surface layer shallower than 50 μm is plastically deformed.

Improvement of tensile residual stress in the lathe machined XM19 ring by the emery paper polishing is shown in Figure 8. For the as-lathe machined condition, the circumferential residual stress is 690 MPa on the surface. By the emery paper polishing, the tensile residual stress rapidly decreases. The residual stress turns to compressive by four polishings and gradually saturates to about −300 MPa. The axial residual stress is 200 MPa on the surface. By the emery paper polishing, the tensile residual stress rapidly decreases. The residual stress turns out to be compressive by the third polishing and

gradually saturates at about −450 MPa. Anyway, the emery paper polishing can easily change the tensile residual stress to the compressive residual stress.

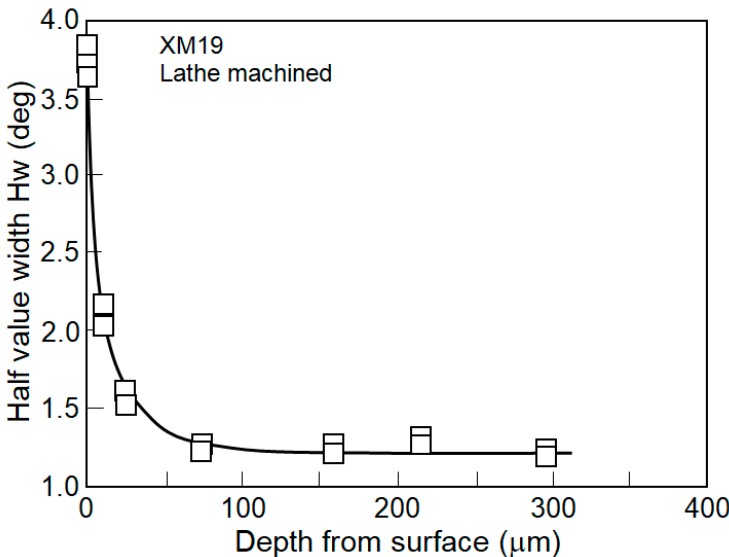

**Figure 7.** Distribution of half value width in the lathe machined XM19 ring.

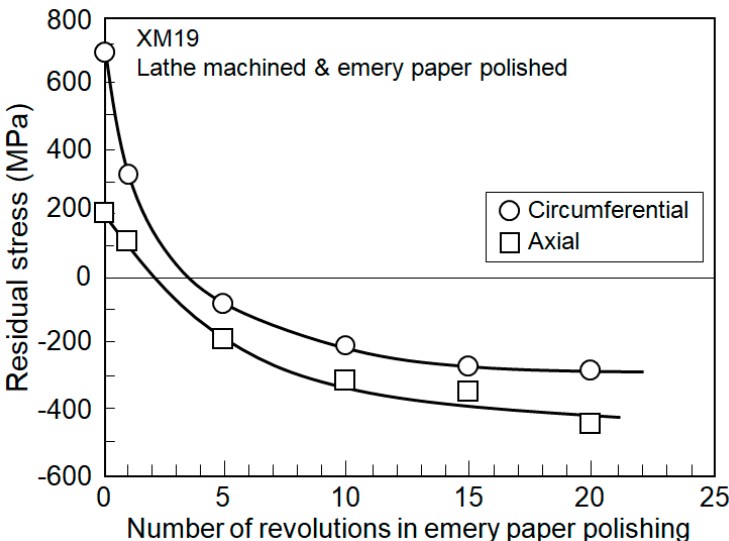

**Figure 8.** Improvement of tensile residual stress by emery paper polishing in XM19.

The residual stress distribution after emery paper polishing is shown in Figure 9. The residual stress on the surface is about −200 MPa. This coincides with that of just emery paper polished surface of solution heat treated material. It turns out to be tensile deeper than 10 μm and takes the compressive at deeper than 50 μm.

The high tensile residual stress surface layer induced by the lathe machining is removed by the emery paper polishing. After the emery paper polishing, the residual stress distribution near the surface becomes that of an emery paper polished solution heat treated material. Thus, the residual stress distribution after emery paper polishing seems to be the combination of the residual stress distribution induced by the lathe machining except the high tensile residual stress surface layer and the residual stress distribution of emery paper polished solution heat treated material.

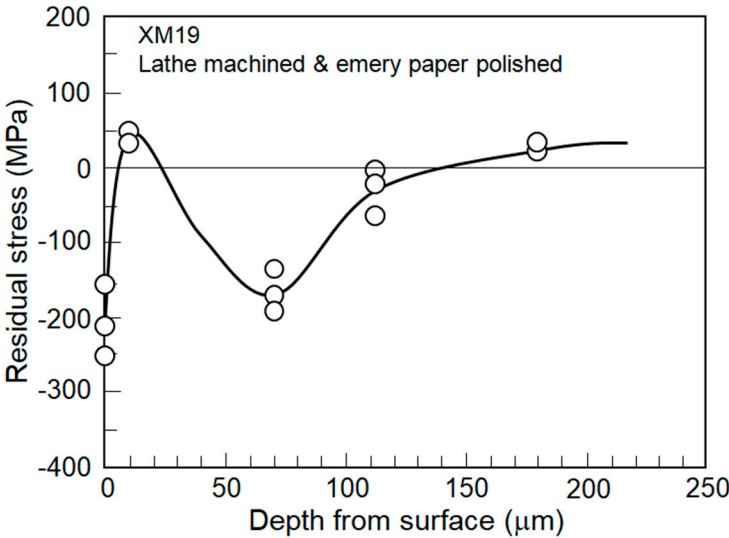

**Figure 9.** Residual stress distribution after emery paper polishing in lathe machined XM19.

The effect of shot peening time on the residual stress in type 304 stainless steel plate is shown in Figure 10. In the shot peening shot size is 0.63 mm in diameter, pressure is 0.54 MPa, and the distance between peening gun nozzle and the plate is about 200 mm. By applying 2.5 s, the residual stress becomes −400 MPa. It gradually decreases with the peening time and saturates after 15 s to about −520 MPa.

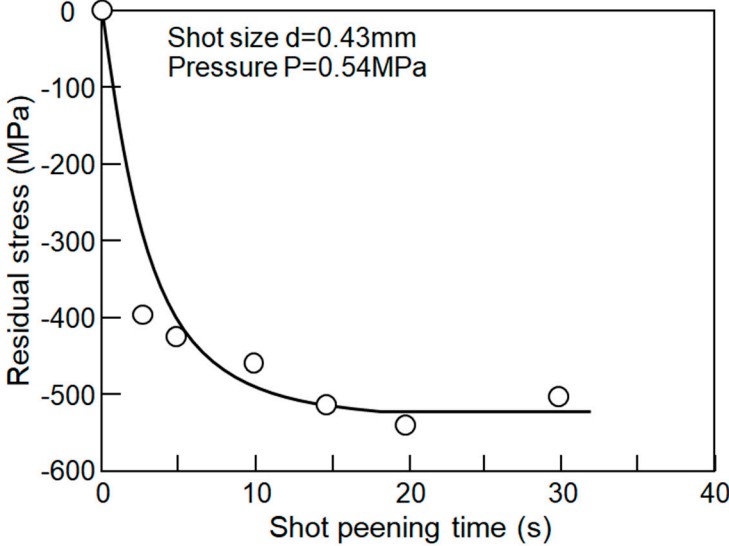

**Figure 10.** Effect of shot peening time on residual stress in type 304 stainless steel.

Residual stress distribution in the shot peened type 304 stainless steel plate is shown in Figure 11. Shot peened time is 2.5, 5, and 10 s. For reference, the residual stress distribution in emery paper polished steel is shown in the Figure 9. The residual stresses induced by the shot peening range −400 MPa and −460 MPa. They decrease with the depth, take the maximum values at about 100 μm, and gradually increase. For balancing with the compressive residual stress, the tensile residual stress is induced for deeper positions.

The residual stress distribution induced by the single shot peening in depth direction has been analyzed using FEM analysis [20]. The results show that the residual stress on the surface is sometimes tensile and gradually decreased with the depth, and took the compressive maximum value at the depth of 250 μm. The residual stresses in surroundings of shot are remarkably small compared with

those at the right below the shot. However, the residual stresses in the surroundings of multi-shot peening have not been clarified.

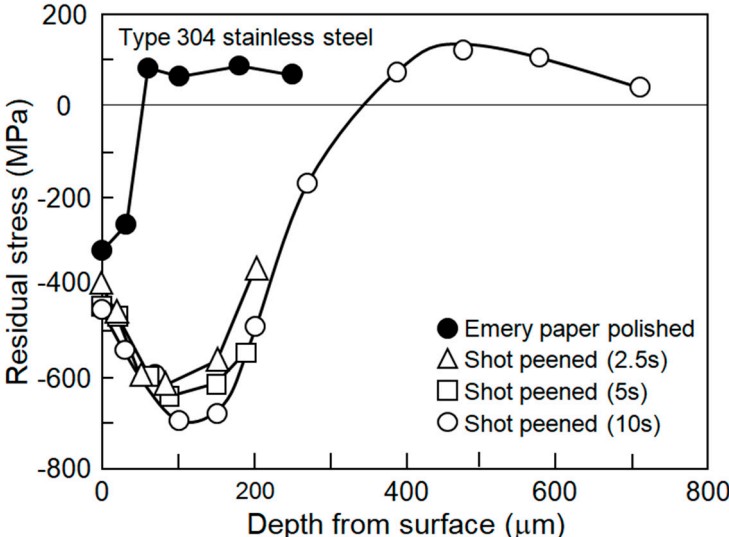

**Figure 11.** Residual stress distribution in shot peened type 304 stainless steel plate.

The residual stress distribution in the subsurface due to the shot peening is schematically shown in Figure 12. Immediately below the surface where the shot hits, the surface layer deforms in the direction perpendicular to the surface. At this moment, the surface layer cannot be deformed in the direction parallel to the surface due to the surrounding constraints. So, the compressive residual stress is generated in the direction parallel to the surface. On the other hand, in the peripheral area hit by the shot, the surface layer deforms following the shape of the shot. This makes the surface morphology like pear surface. The part protruding from the surface is bent by the shots on both sides and then the compressive residual stress seems to be not so high. Extremely, the residual stress may become tensile. Thus, it is possible to greatly improve the residual stress formed by shot peening by removing the protruding parts with low compressive residual stress with emery paper.

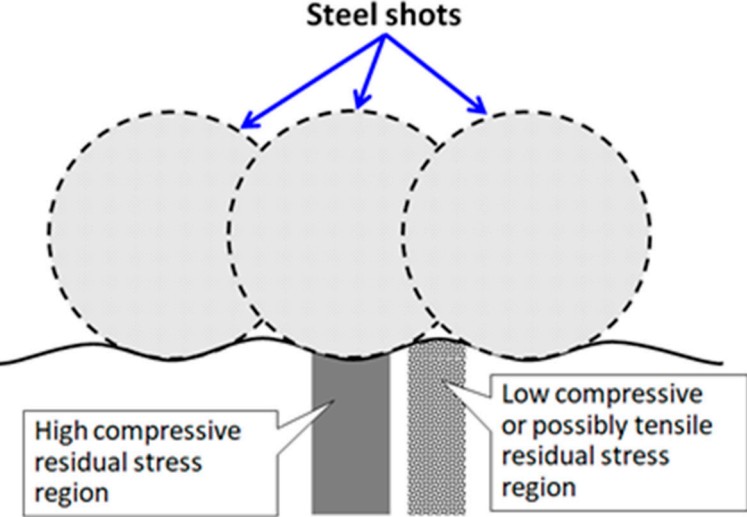

**Figure 12.** Residual stress distribution in subsurface due to shot peening.

Figure 13 shows the change in the residual stress formed by shot peening depending on the number of times of polishing with #240 emery paper. In the case of shot peening time of 10 s, the surface

residual stress was improved from −400 MPa to −700 MPa only by polishing 20 times. In the case of shot peening time of 20 s, the surface residual stress was improved from −560 MPa to −750 MPa only by polishing 20 times. According to the analytical results of the stress field in the shot peening, it is thought that a higher compressive stress is formed due to high shear stress a little far away from the surface.

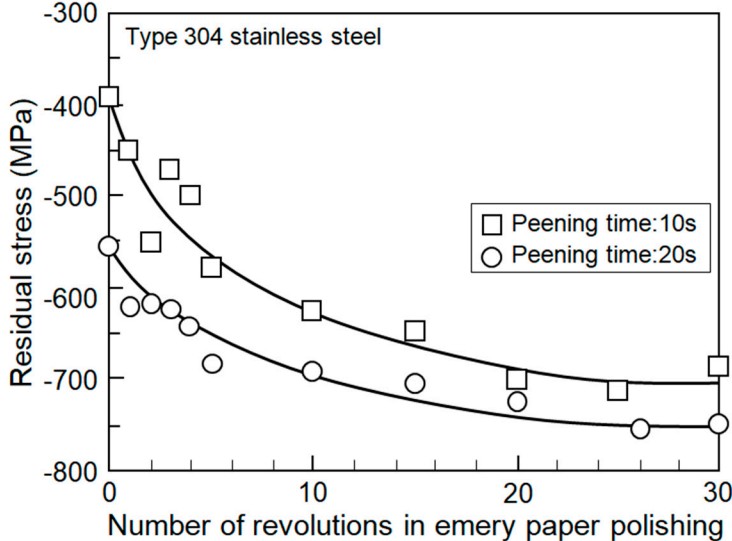

**Figure 13.** Improvement of residual stress by emery paper polishing in shot peened type 304 stainless steel.

Surface morphology of shot peened and followed by emery paper polishing in type 304 stainless steel is shown in Figure 14. Since both sides are as shot peened, the surface is a pear-like and has a large roughness, and the photograph looks black. This pear-like surface was removed and polished using #240 emery paper. As a result, the 20 times polished surface gives a metallic luster, and the roughness is much smaller. On the 10 times polished surface, many dents can be observed. These dents are formed by the impacts of the shots.

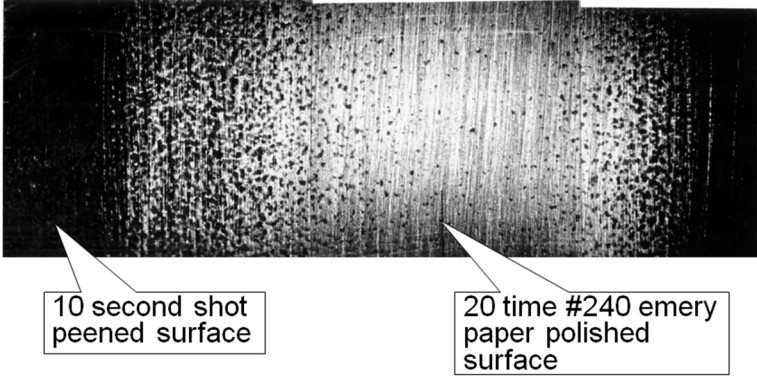

**Figure 14.** Surface morphology of shot peened and followed by emery paper polishing in type 304 stainless steel.

### 3.2. Fatigue Tests of Various Preliminary Worked-Steel

S–N curves for electro-polished, emery paper polished, shot peened, and shot peened plus emery paper polished steels in air at 288 °C are shown in Figure 15. For the stress amplitude higher than 250 MPa, the effects of compressive residual stress and work hardening due to the plastic deformation are attenuated by repeated stress. Thus, the effect of surface working decreases, and the fatigue lives

are almost the same in spite of the preliminary surface working. However, the effect of preliminary surface working is remarkable for the stress amplitude lower than 250 MPa. While the $10^6$ cycle fatigue strength of the electro-polished material is 162 MPa, that of emery paper polished material is 167 MPa. As shown in Figure 11, the compressive residual stress on the surface due to the emery paper polishing is high at −300 MPa, but the plastically deformed layer with compressive residual stress is as shallow as about 20–30 μm. This is the reason why the fatigue strength was improved by only about 10 MPa.

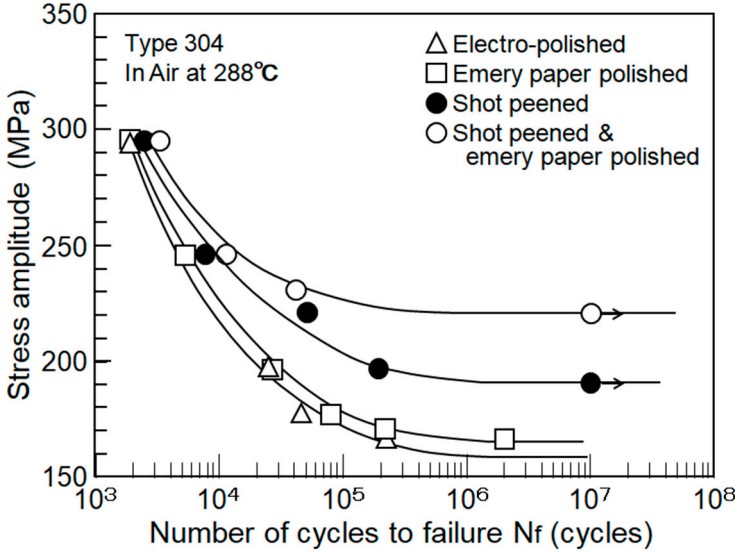

**Figure 15.** S–N curves of various preliminary worked type 304 stainless steels at 288 °C in air.

In the shot peened material, the fatigue strength is 191 MPa. This increase is caused by the large compressive residual stress on the surface, the remarkable plastic-hardening on the surface layer, and the deep plastically deformed region.

Fatigue strength of specimens polished with emery paper after shot peening, which is denoted as SE specimen, is 221 MPa. In SE specimens, the compressive residual stress is about 300 MPa larger than that of a shot peened specimen, while the work-hardening is almost the same. In spite of this, the fatigue strength is improved about 30 MPa. This seems to be due to the improvements of surface roughness and the local residual stress. In SE specimens, the pear-like surface formed by the shot peening is removed by the emery paper polishing and the final surface becomes smooth. However, the fatigue strength improvement of 15% could not be expected by the surface smoothing. It is suggested that the local residual stress affected the fatigue strength. As shown in Figure 14, the residual stress at the part protruding from the surface is not highly compressive. This is obvious from the fact that the surface residual stress is improved about 300 MPa by the emery paper polishing. Conversely, this means that the residual stress at the part protruding from the surface would be −100 MPa. It is considered that the removal of this protruding portion with a relatively low compressive residual stress is effective in significantly improving fatigue strength.

## 4. Conclusions

The effects of turning and shot peening on the residual stress distributions in type 304 stainless steel measured by the X-ray diffraction method, and their improvements by the emery paper polishing, were examined. The conclusions obtained in this study would be summarized as follows:

(1)　In lathe machined type 304 stainless steel round bar, the residual stresses in the axial direction were almost compressive, and the residual stresses in the circumferential direction were very highly tensile. The difference between the mild steel and the austenitic stainless steel seems to be the very high work-hardening rate and the very low heat conductive efficiency of the austenitic stainless steel.

(2)　The very high tensile residual stresses were easily improved to the compressive residual stresses by the emery paper polishing. The number of polishing required to change the tensile residual stress to the compressive residual stress is at most 20 times.

(3)　In the shot peened surface, the residual stress is about −400–500 MPa on the surface and takes the maximum value of about −700 MPa at the depth of 150 μm.

(4)　By the shot peening, the surface becomes pear-like. Twenty polishings using #240 emery paper makes the surface flat, and the residual stress is improved up to −700 MPa.

(5)　$10^6$ cycle fatigue strength of the electro-polished type 304 stainless steel is 162 MPa, and that of emery paper polished material is 167 MPa. The fatigue strength of shot peened steel is 190 MPa, and the improvement is no more than 30 MPa. When the shot peened surface is polished with the emery paper, the fatigue strength reaches 220 MPa and is highly improved by 60 MPa.

(6)　The emery paper polishing is very simple and a very effective process for the improvement of the residual stresses and the fatigue strength.

**Funding:** This research received no external funding.

**Conflicts of Interest:** The authors declare no conflict of interest.

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
