# Peer review of "Residual Stresses Induced by Surface Working and Their Improvement by Emery Paper Polishing"

_qubs, doi:10.3390/qubs4020021_

Round 1

Reviewer 1 Report

A very interesting article, it touches and describes the significant problems of surface treatment of machine and device elements and the method of its assessment by measuring residual stress. The author has presented a lot of interesting data of great practical importance.
However, some passages seem to need to be improved or supplemented.

Table 3. - The depth of X-ray penetration, and hence the thickness of the layer from which the results were obtained should be given.

Graphs - On the graphs showing experimental data for stress changes, error bars would be appreciated

Line 131-137.

In the experiment described above it could be that residual stresses were compressive in both directions after turning. However, very often there is a situation when, also for carbon steel, residual stresses have a tensile character in one or both directions. It depends on the machining parameters used and cannot be generalized.

Line 207 – It should be 100 µm

Line 185 – 212 - What was the procedure for measuring the stress at depth (by electrochemical etching or…?).

Line 281 and 282 - This is true but it adds nothing, you can also measure by X-ray. It is more often used as a cheaper one and does not require such complicated apparatus as the neutron diffraction method. Depth distribution is obtained by stripping the layers by electrochemical etching.

Line 284 – 288 - The sentences are true, but for both materials with a properly selected set of machining parameters for each of them, similar residual stress distributions can be obtained. So these statements do not add anything new

Author Response

Comment 1:

Table 3. - The depth of X-ray penetration, and hence the thickness of the layer from which the results were obtained should be given.

Graphs - On the graphs showing experimental data for stress changes, error bars would be appreciated.

*Answer

The incident X-ray beam is decayed by the absorption. The penetration depth is defined as the depth at which the intensity of X-rays entering the material is reduced to 1 / e of the incident X-ray intensity on the surface. The penetration depth in the austenitic stainless steel using Cr-Kb is estimated as 7.1 mm.

Comment 2:

Line 131-137.

In the experiment described above it could be that residual stresses were compressive in both directions after turning. However, very often there is a situation when, also for carbon steel, residual stresses have a tensile character in one or both directions. It depends on the machining parameters used and cannot be generalized.

*Answer

In Japan many researchers studied the residual stresses in severally worked material. In case of turning most of researchers obtained the compressive stresses both in the axial and circumferential directions. Surely they depend on the turning conditions. So that the references showing the results of the residual stress measurements in the variously worked carbon steel are added in the sentence and the references.

Comment 3:

Line 207 – It should be 100 µm

*Answer

“100 mm” is wrong and is modified as 100 μm. 

Comment 4:

Line 185 – 212 - What was the procedure for measuring the stress at depth (by electrochemical etching or…?).

*Answer

The residual stress distributions in the sub-surface were measured by the sequential polishing method using electrochemical etching liquid.

The above sentence is added in section of 2.3 Residual stress measurement.

Comment 5:

Line 281 and 282 - This is true but it adds nothing, you can also measure by X-ray. It is more often used as a cheaper one and does not require such complicated apparatus as the neutron diffraction method. Depth distribution is obtained by stripping the layers by electrochemical etching.

*Answer

Line 281 and 282 is a typo.

Modify as follows;

The effects of turning and shot peening on the residual stress distributions in type 304 stainless steel measured by the neutron diffraction, and their improvements by the emery paper polishing were examined.

Comment 6:

Line 284 – 288 - The sentences are true, but for both materials with a properly selected set of machining parameters for each of them, similar residual stress distributions can be obtained. So these statements do not add anything new

*Answer

So far the residual stresses were measured mainly in the carbon steels not in the stainless steels. As mentioned in the paper the circumferential residual stresses were tensile. This is different from the results obtained in the carbon steel. The mechanism of the difference is clearly explained. Thus conclusion (1) has to be mentioned, I think.

Reviewer 2 Report

The paper is within the scope of the journal and deals with investigating the effect of surface working on residual stress state and fatigue properties of steels 304 and XM19. The topic is of high interest; however, the manuscript contains several weak points. Therefore, a substantial revision is required

Comments:

  1. Even the most of figures are very accurate represented; the manuscript text contains a lot of typos and grammar errors. Therefore a substantial English revision is required.
  2. In the introduction the state of the art and motivation for the study is not clear. The literature review is not comprehensive enough. Some statements are without references, e.g., “residual stress affect the tensile strength”. Why it is? In my opinion, it is mostly not the case. Moreover, other statements require clear explanations and supporting references. I suggest also to include the literature review regarding other residual stress engineering techniques such as shot peening, laser shock peening etc. and to explain their potential for the fatigue improvement.
  3. “2.3. Residual stress measurement”. As residual stresses do not physically exist and it is only a concept, it is not possible to measure residual stresses. You can only measure strains! Please correct the titles and sentences in the manuscript.
  4. Now the authors only described the obtained the results. Therefore, it is a lab report and not a journal paper. The authors have to perform a comprehensive discussion of the obtained results and to compare the results with the results available in the literature.
  5. I found only one references in the chapter “results and discussion”: “The residual stresses … has been analysed using FEM”. However, it is not clear how it was done.

Author Response

Comment 1:

In the introduction the state of the art and motivation for the study is not clear. The literature review is not comprehensive enough. Some statements are without references, e.g., “residual stress affect the tensile strength”. Why it is? In my opinion, it is mostly not the case. Moreover, other statements require clear explanations and supporting references. I suggest also to include the literature review regarding other residual stress engineering techniques such as shot peening, laser shock peening etc. and to explain their potential for the fatigue improvement.

*Answer

Isn't it a well-known fact that residual stress affects tensile strength?

The representative references are added and the former part of introduction is modified. Could you check the revised manuscript.

Comment 2:

“2.3. Residual stress measurement”. As residual stresses do not physically exist and it is only a concept, it is not possible to measure residual stresses. You can only measure strains! Please correct the titles and sentences in the manuscript.

*Answer

Title of this Special Issue is “Residual Stress and Texture”.

When the load, L, is subjected to the material the cross sectional area of which is A, the stress s is generated as s=L/A, and at the same time the strain e is generated as e=s/E (E:Young’s modulus) according the elastic deformation.

When the material is plastically deformed, the elastic strain and the plastic strain coexist in the material. When the material is unloaded, the residual stresses are generated and their values are calculated through Hook’s law using the residual strain. In order to evaluate the fatigue strength, necessary is the residual stress but not the residual strain. So, it is not necessary to change the title.

Comment 3:

Now the authors only described the obtained the results. Therefore, it is a lab report and not a journal paper. The authors have to perform a comprehensive discussion of the obtained results and to compare the results with the results available in the literature.

*Answer

The author describes the difference with the residual stress introduced in the carbon steel and the mechanism of its difference is suggested. Would you please check the manuscript.

Comment 4:

I found only one references in the chapter “results and discussion”: “The residual stresses … has been analysed using FEM”. However, it is not clear how it was done.

*Answer

I think it is not necessary to explain the details of FEM analysis in this manuscript. Would you please check reference No.20.

Round 2

Reviewer 2 Report

The authors addressed the comments appropriately. The manuscript can be accepted for the publication in the Journal.

Author Response

Thank you for your kind work